# Optimal Dataset Design for Nurture-then-Nature Teaching

## ABSTRACT

Designing an optimal dataset to teach a target concept to a learner has been a well-studied problem in Machine Learning. Prior works have mostly focused on unconstrained single-phase teaching, where the learner learns solely under the guidance of a helpful teacher who can provide any number of examples. In this work, we introduce a more realistic two-phase framework called "Nurture-then-Nature" where the learner first learns under the guidance of a teacher in the 'Nurture' phase, followed by an i.i.d. learning phase from 'Nature'. Importantly, the teacher is constrained to provide a dataset of size up to $B$ and is required to minimize the final error of the learner. We study this problem in the 'instance-agnostic' and 'instance-aware' settings and provide efficient teaching algorithms for each of them. We provide theoretical guarantees and experimental results to support our findings.

## 1 INTRODUCTION

The problem of designing an optimal dataset to teach a target concept $h^* : \mathcal{X} \to \mathcal{Y}$ to a learner, also known as Machine Teaching, has been a long-studied problem Goldman & Kearns (1995); Liu & Zhu (2016); Zhang et al. (2016). Prior works on Optimal Teaching have mainly focused on a single-phase learning setting where the student learner solely learns under the guidance of the teacher who has unconstrained teaching budget Goldman & Kearns (1995); Zhang et al. (2016); Kumar et al. (2021); Liu & Zhu (2016). However, in many practical scenarios, the teacher may only have a limited budget of teaching lessons that it can provide to the learner. For example, consider a university curriculum setting where a teacher has to teach a concept, say how to identify a disease from an MRI scan to a student(see Figure 1) but it can only teach a limited number of lessons (dataset $D_T : |D_T| \leq B$) to them before they graduate from the program. This first phase of learning which takes place under the guidance of the teacher is called the "Nurture" phase. Since *"Teaching Dimension"*(TD) Goldman & Kearns (1995) is the smallest possible dataset to teach a concept, the teacher will not be able to teach completely, if the budget is less than *TD* Goldman & Kearns (1995).

However, from the student's perspective, learning does not stop after graduating from the university. Rather, they transition to a "Nature" learning phase and continue to learn about the target concept by receiving an i.i.d. dataset $D_E$ from the nature/environment. For example in Figure 1, the student keeps learning about disease identification from i.i.d. MRI scans drawn from a digital library with a hope to master the concept over time. We call this two-phase learning setting **"Nurture-then-Nature"(NtN)** learning. The goal of a good teacher is to design an "optimal" dataset $D_T^*$ to minimize students' error at the end of nature phase in NtN learning. To study this further, we ask the following question:

*What is an optimal teaching demonstration to minimize the error in NtN setting?*

We study this problem and make the following contributions: 1.) We propose a novel mathematical framework of "Nurture-then-Nature" learning for studying budget-constrained teaching where the goal of the teacher is to minimize final error of the student. 2.) We study the problem under two levels of knowledge by the teacher and propose teaching algorithms for each of them:

1. In **Instance Agnostic** setting, we consider a teacher who does not know the environment distribution $P$ and is required to teach instances with any $P$. Our efficient teacher constructs an optimal teaching set to simplify the complexity of learner's version space at the end of 'Nature' phase thereby making it easy for them to learn from i.i.d. sample in 'Nature' phase.

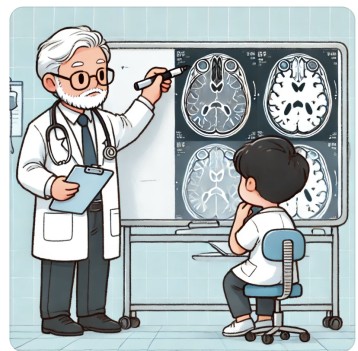 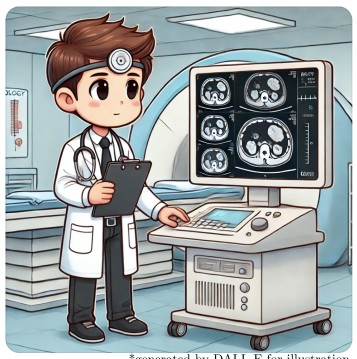

*generated by DALL-E for illustration.

Nurture Learning: under a Teacher.     Nature Learning: from i.i.d. dataset.

Figure 1: An illustrative example of Nurture-then-Nature learning in medical teaching domain. In the 'nurture' phase, the student learns to identify a brain disease from MRI scans under the guidance of a teacher. This is followed by the 'nature' phase where the student continues learning using an i.i.d. sample from a digital database.

2. In **Instance Aware** setting, the teacher knows $P$ and is required to be competitive with the instance optimal teaching set. We propose an algorithm using datamodels Ilyas et al. (2022) to exactly solve this problem under linear assumption. Unlike (a), this method works for any learner that satisfies linear risk assumption and extends to non-linear datamodels as well.

We present both theoretical and experimental results to validate the effectiveness of our algorithms in both settings and compare their performance to a simulated baseline algorithm.

## 2 RELATED WORKS

Machine Teaching has been a well-studied problem in the literature Shinohara & Miyano (1991); Goldman & Kearns (1995); Zhu (2015). Past works have studied optimal teaching in various learning settings ranging from supervised learning Goldman & Kearns (1995); Liu & Zhu (2016); Kumar et al. (2021); Bharti et al. (2024) to online/active learning Zhang et al. (2016); Peltola et al. (2019) to sequential decision making and reinforcement learning Brown & Niekum (2019); Tschiatschek et al. (2019); Zhang et al. (2021). However, most of these works have focused on unconstrained teaching setting where the teacher is free to design and teach a dataset of any arbitrary size which may not be possible under real-world constraints. Our work studies budget-constrained teaching in a two-phase supervised learning setting where the teacher can only provide a dataset up to a fixed size.

Some recent works have considered other forms of constraints that are distinct from our budget constraints like time constraint Filho et al. (2023), preference constraint Tschiatschek et al. (2019). The most relevant work to ours is the budget-constrained teaching problem examined by Kobayashi & Shinohara (2009). However, the authors have only considered a single-phase teaching setting of Goldman & Kearns (1995) where the goal is to minimize the learner's error at the end of the teaching phase. Moreover, their algorithm and analysis is very specialized to distribution-independent teaching of a class of monomials Kearns et al. (1994). On the other hand, our framework is much more general with a clearly different goal. Furthermore, our teaching algorithms can handle infinite hypothesis classes like linear/polynomial classifiers through VC reduction and linear datamodel connections.

Optimal teaching has been shown to be a hard bilevel optimization problem Goldman & Kearns (1995); Zhu et al. (2018) which limits its practical utility. The main difficulty often lies in estimating the risk of the learner as a function of the dataset. Naive methods require simulating a learner to estimate the risk on different independent datasets making it a challenging task. However, recent works like linear datamodels Ilyas et al. (2022) have taken a function approximation approach and have shown that risk can be well approximated by a linear function of the dataset in many real-world problems. Prior works have utilized this connection to detect backdoor attacks Khaddaj et al. (2023), forget training data using machine unlearning Georgiev et al. (2024) and to select good datasets for

training large models Engstrom et al. (2024) which aligns closely with the objective of single-phase machine teaching setting, which is clearly different from our budget-constrained Nature-then-Nature teaching.

# 3 PROBLEM FORMULATION

## 3.1 THE LEARNER AND THE ENVIRONMENT

Consider a predictive modeling task from an input space $\mathcal{X}$ to an output space $\mathcal{Y}$ defined by a joint distribution $P$ over $\mathcal{X} \times \mathcal{Y}$. During learning, a learner receives a dataset $D = \{(x_i, y_i)\}|_{i \in [n]} \subseteq (\mathcal{X} \times \mathcal{Y})^n$ and tries to learn a good predictive model that does well on future data from $P$.

An Empirical Risk Minimization (ERM) Shalev-Shwartz & Ben-David (2014); Mohri et al. (2018) learner/student $\mathcal{A}$ starts with a hypothesis class $\mathcal{H}$ and minimizes empirical risk with respect to a loss $\ell : \mathcal{Y} \times \mathcal{Y} \to \mathbb{R}_{\geq 0}$ on the training dataset $D$,

$$\mathcal{A}(D; \mathcal{H}) = \arg\min_{h \in \mathcal{H}} \frac{1}{n} \sum_{(x_i, y_i) \in D} \ell(h(x_i), y_i) \tag{1}$$

where, $\ell$ is the loss function. It eventually aims to learn a hypothesis with the smallest risk $R_P(h) = \mathbb{E}_{(x,y) \sim P}[\ell(h(x), y)]$. We make the following simplifying assumption on the realizability of the environment which has been well used in literature Goldman & Kearns (1995); Liu & Zhu (2016).

**Definition 1** (Realizability & Version Space Learner). *An environment is said to be realizable if $\exists h^* \in \mathcal{H}$ such that $P = P_X \cdot P_{Y|X}$ and $P_{Y|X}(Y = h^*(X)) = 1$. Under realizability, an ERM learner that minimizes the risk w.r.t. $0 - 1$ loss and maintains the entire subset of ERM hypotheses is called a version space(VS) learner.*

**Remark 1.** *We note that the output of learning $\mathcal{A}(D; \mathcal{H})$ can be a single hypothesis or a subset of them, depending on the learner.*

## 3.2 THE TEACHER

There is a helpful teacher who is required to teach a target hypothesis $h^* \in \mathcal{H}$ to the learner. The teacher knows $h^*$ but can only provide a dataset of size up to budget $B \in \mathbb{Z}^+$ to the learner before they graduate. The teacher will not be able to teach $h^*$ completely if $B$ is less than the $TD$. However, after graduating, the learner keeps learning about $h^*$ using i.i.d. sample from the environment $P$.

We consider two teaching settings based on different levels of knowledge of the teacher:

1. **Instance agnostic setting**: In this setting, the teacher does not know the underlying $P_X$ and has to teach a learner in instance agnostic way, i.e. the teaching should work for any $P_X$.

2. **Instance aware setting**: In this setting, the teacher knows the underlying distribution of the environment $P_X$ and has to be competitive wrt to instance optimal solution.

Next, we define the interaction of the learner with the teacher and the environment.

## 3.3 THE NURTURE-THEN-NATURE SETTING

The learning process of the version space learner in the NtN setting is split into two phases:

**Phase I - The Nurture Phase:** In this phase, the learner learns under the guidance of the teacher. It receives a dataset $D_T$ from the teacher and learns a version space of hypothesis $\mathcal{V}(D_T; \mathcal{H})$ consistent with $D_T$ given as,

$$\mathcal{V}_1 := \mathcal{V}(D_T; \mathcal{H}) = \{h \in \mathcal{H} \mid h(x_i) = y_i, \forall (x_i, y_i) \in D_T\}. \tag{2}$$

**Phase II - The Nature Phase:** The nurture phase is followed by the *nature/i.i.d. learning phase* where the learner starts with the surviving version space $\mathcal{V}_1$ from previous phase and continues to learn about $h^*$ by receiving a i.i.d. dataset $D_E \sim P^n$ of size $n$ from the environment distribution $P$. It then learns a version space $\mathcal{V}(D_E; \mathcal{V}_1)$ consistent with $D_E$ on $\mathcal{V}_1$, i.e.,

$$\mathcal{V}_2 := \mathcal{V}(D_E; \mathcal{V}_1) = \{h \in \mathcal{V}_1 : h(x_i) = y_i, \forall (x_i, y_i) \in D_E\}.$$

At the end of this phase, the learner hopes to have learned $h^*$ with as small a risk as possible.

**Remark 2.** *We make the following remarks on the two phases:*

1. *With budget $B \geq TD$, this phase captures the standard unconstrained teaching problem Goldman & Kearns (1995). However, with a budget $B < TD$, the teacher can only teach $h^*$ partially to the learner, leading to a very different teaching problem.*

2. *We note that in the Nature phase, the learner learns using a hypothesis class $\mathcal{V}_1$ that has been simplified from $\mathcal{H}$ by the teaching set $D_T$ provided by the teacher. In effect, the teacher controls the complexity of learning from Nature by simplifying $\mathcal{V}_1$ using $D_T$.*

## 4 INSTANCE AGNOSTIC TEACHING SETTING

In instance agnostic setting, the teacher does not know the environment distribution $P_{\mathcal{X}}$ and aims to minimize a high probability instance agnostic objective defined as follows:

**Teaching objective**: Given an instance $(\mathcal{X}, \mathcal{Y}, P_X, h^*, \mathcal{H}, \delta, n, B)$, the instance agnostic teaching objective is defined as,

$$D_T^* \leftarrow \underset{\epsilon, D_T : |D_T| \leq B}{\arg\min} \ \epsilon$$

$$\text{s.t. } \forall P, \quad \mathbb{P}_{D_E \sim P^n} \left( \max_{h \in \mathcal{V}(D_E; \mathcal{V}(D_T; \mathcal{H}))} R_P(h) \leq \epsilon \right) \geq 1 - \delta \tag{3}$$

**Remark 3.** *We make the following remarks: 1.) The teacher does not know $P_X$, it has to ensure that learner succeeds in any $P$. 2.) The teacher influences learner's performance through a budget constrained dataset $D_T : |D_T| \leq B$ that reduces $\mathcal{H}$ to $\mathcal{V}(D_T; \mathcal{H})$.*

Note that the feasibility constraint in 3 requires to satisfy $(n, \delta)$ PAC guarantee for any $P$. Prior works in PAC-learning Haussler et al. (1994); Vapnik (1992) provides the following instance agnostic bound on error of version space learner that helps to simplify the problem objective.

**(PAC-guarantee)** : $\forall P, \forall n \in \mathbb{N}, \delta \in (0, 1), \quad \text{if } D \overset{iid}{\sim} P^n,$
$$w.p. \geq 1 - \delta, \quad \max_{h \in R(\mathcal{V}(D; \mathcal{H}))} R(h) \leq \epsilon(n, \delta, \mathcal{H}) \tag{4}$$

where, $R(h)$ is the risk with respect to the $0 - 1$ loss. For a hypothesis class $\mathcal{H}$ and a fixed $(n, \delta)$, PAC-guarantee satisfies 4 with $\epsilon(n, \delta, \mathcal{H}) = O(\frac{1}{n} \cdot (d(\mathcal{H}) + \log(\frac{1}{\delta})))$ where $d(\mathcal{H})$ is the VC dimension of $\mathcal{H}$. Later, Hanneke (2016) also proved that this guarantee is optimal with respect to $d(\mathcal{V}(\mathcal{H}))$.

Note that the feasibility constraints of Equation 3 is nothing but PAC-guarantee with surviving version space $\mathcal{V}_1 = \mathcal{V}(D_T, \mathcal{H})$ as the hypothesis space. This reduces our teaching objective in 3 to:

$$D_T^* \leftarrow \underset{D_T : |D_T| \leq B}{\arg\min} \ \frac{1}{n} \left( d(\mathcal{V}(D_T; \mathcal{H})) + \log\left(\frac{1}{\delta}\right) \right). \tag{5}$$

Since $(n, \delta)$ are fixed, we essentially need to minimize the VC of the version space $\mathcal{V}(D_T; \mathcal{H})$ maximally under budget constraint $B$ leading to the following theorem on the teaching algorithm.

**Theorem 1.** *A teaching algorithm that, by teaching using dataset $D_T^*$, optimally reduces the VC dimension of the version space $\mathcal{V}(D_T; \mathcal{H})$ surviving at the end of the Nurture phase solves the instance-agnostic NtN teaching problem optimally.*

Computing VC is tractable for hypothesis classes like axis-aligned rectangles, linear classifiers, polynomial classifiers Mohri et al. (2018); Shalev-Shwartz & Ben-David (2014), however, in general this is a NP-hard problem Shinohara (1995); Manurangsi & Rubinstein (2017); Manurangsi (2022). Since, optimally reducing VC is at least as hard as computing it, we cannot hope to reduce VC of general hypothesis classes efficiently. Instead, we focus on optimally reducing the VC dimension for tractable hypothesis classes under finite teaching budget.

We begin with one of the simplest hypothesis class, a finite binary hypothesis class Goldman & Kearns (1995) and then extend our analysis to several other hypothesis classes.

## 4.1 FINITE BINARY HYPOTHESIS CLASS.

A finite binary hypothesis class consists of a set of hypothesis, each mapping a finite input space $\mathcal{X}$ to binary labels $\{0, 1\}$, i.e., $\mathcal{H} \subseteq 2^{\mathcal{X}}$. We know that computing VC of $\mathcal{H}$ takes $\Theta(n^{\log(n)})$ time Papadimitriou & Yannakakis (1996); Manurangsi & Rubinstein (2017) and is NP-hard. This eventually makes optimizing for VC an NP-hard problem as well. Hence, we further upper bound VC by the size of the hypothesis class and aim to minimize that instead.

Given a budget $B$, the teacher aims to find teaching set $D_T \subseteq \mathcal{X}, |D_T| \leq B$, that reduces the size of version space $\mathcal{V}(D_T; \mathcal{H})$ maximally as follows:

$$D_T^* \leftarrow \arg \min_{D_T : |D_T| \leq B} |\mathcal{V}(D_T; \mathcal{H})|. \tag{6}$$

It turns out that even this problem is NP-hard since it's equivalent to another NP-hard problem called *Budgeted Maximum Coverage Problem* Khuller et al. (1999). However, there exists an efficient algorithm to solve this problem approximately leading to the following theorem.

**Theorem 2.** *There exists an efficient algorithm that reduces the version space size of finite hypothesis class up to an approximation ratio of $1 - \frac{1}{e}$.*

The algorithm and the proof of theorem can be found in the appendix. Next, we study another classic hypothesis class considered in literature, the axis aligned rectangle hypothesis class Goldman & Kearns (1995).

## 4.2 AXIS-ALIGNED RECTANGLES ON $\mathbb{Z}^2$ GRID

This class consists of all axis-aligned rectangles in $\mathbb{Z}^2$ space. A hypothesis $h \in \mathcal{H}$ is defined by the two opposite corners $(x_{\min}, y_{\min}), (x_{\max}, y_{\max}) \in \mathbb{Z}^2$ and it produces the following classifier:

$$h((x, y)) = 2 \cdot \mathbb{1}[x_{\min} \leq x \leq x_{\max} \wedge y_{\min} \leq y \leq y_{\max}] - 1.$$

We recall that VC of this class $d_{VC}(\mathcal{H}) = 4$ Mohri et al. (2018), and, the TD for teaching any $h \in \mathcal{H}$ is 6 Goldman & Kearns (1995). For our NtN setting, we focus on non-trivial cases with $B < TD$.

**Theorem 3** (Optimal VC reduction for axis-aligned rectangles.)**.** *The VC dimension of axis-aligned rectangles in $\mathbb{Z}^2$ can be optimally reduced as follows:*

| Budget $B$ | 1 | 2 | 3 | 4 | 5 | $\geq 6$ |
|---|---|---|---|---|---|---|
| min VC | 4 | 3 | 2 | 2 | 1 | 0 |

Table 1: Minimum VC achievable by $B$-budgeted teaching on axis-aligned rectangle class in $\mathbb{Z}^2$.

We refer the readers to appendix for a complete proof. Next, we consider two popular hypothesis classes, a homogeneous linear and a polynomial class.

## 4.3 LINEAR HYPOTHESIS CLASSIFIERS IN $\mathbb{R}^d$

Consider teaching a family of homogeneous linear binary classifiers in $\mathcal{H} = \mathbb{R}^d$. Given a $w \in \mathbb{R}^d$, it induces a linear classifier of form,

$$h_w(x) = 2 \cdot \mathbb{1}[w^\top x \geq 0] - 1.$$

Prior works have studied optimal teaching of linear decision boundaries in unconstrained setting and have shown that $TD = d + 1$ for perceptron learner Kumar et al. (2021) and $TD = 2$ for max-margin learner Liu & Zhu (2016). We also know that the VC-dimension of linear class $\mathcal{H}$ is $d$ Mohri et al. (2018) and address the following question:

*"How to optimally reduce the VC-dimension $\mathcal{V}(D_T; \mathcal{H})$ using a constrained teaching set $|D_T| \leq B$?"*.

To do so, we characterize the version space in terms of a polyhedral cone and prove that minimizing VC eventually requires reducing the ambient dimensionality of the version space as stated in

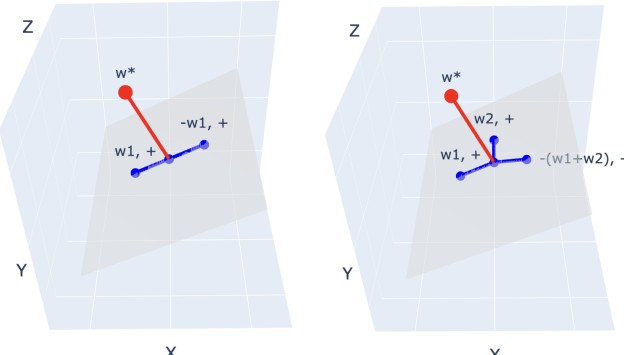

Figure 2: Teaching dataset produced by our optimal VC reduction algorithm for teaching $w^* \in \mathbb{R}^3$ with $B = 2$ and $B = 3$ kills a one and two-dimensional subspace of $(w^*)^\perp$ respectively.

Lemma 10,11 of the appendix. We then provide an algorithm to compute the optimal dataset that essentially works by killing the $B - 1$ orthogonal subspace of $w^*$ on a budget $B$ leading to the theorem stated next.

**Theorem 4.** *There exists an algorithm that $\forall B \leq d+1$, optimally reduces VC of the linear class to $d - B+1$ and optimal teaching set is given as $D_T^B = \{(v_1, +1), \ldots, (v_{B-1}, +1), (-\sum_{i \in [B-1]} v_i, +1)\}$, where, $\{v_1, \ldots, v_{B-1}\}$ is a B-basis of $w^{*\perp}$ subspace.*

Our analysis involves a novel way of characterizing VC of linear version spaces and we utilize it to provide guarantees on optimally reducing VC using a teaching dataset. The complete proof is deferred to the appendix. Next, we consider a polynomial hypothesis class in $\mathbb{R}^d$.

### 4.4 Polynomial Hypothesis Classifiers in $\mathbb{R}^d$

Let $\mathcal{H}$ be hypothesis class of $k$-degree polynomial classifiers in $\mathbb{R}^d$, given by,

$$\mathcal{H} = \{h \mid h(\mathbf{x}) = \mathbb{1}[\sum_{|\alpha| \leq k} w_\alpha \mathbf{x}^\alpha \geq 0], \ \alpha \in \mathbb{N}^d\}$$

and let $\phi : \mathbb{R}^d \to \mathbb{R}^l$ denote the feature mapping for the corresponding Kernel Hilbert space. We know that $l = \binom{d+k-1}{k}$ and the bases feature functions is given by $\mathcal{B} = \left\{\mathbf{x}^\alpha = x_1^{\alpha_1} x_2^{\alpha_2} \ldots x_d^{\alpha_d} \mid |\alpha| \leq k, \sum_j \alpha_j = k, \ \alpha \in \mathbb{N}^d\right\}$. Furthermore, any hypothesis $h \in \mathcal{H}$ can be represented by a parameter $w^* \in \mathbb{R}^l$ in the basis of the Hilbert space.

The polynomial classifier is linear in the $\phi$ feature space and the teacher aims to minimize the VC of the corresponding version space. However, unlike the linear model, each of the teaching input vectors in feature space must be realizable under the feature function $\phi$ on some $x \in \mathbb{R}^d$.

Assuming the feature function is rich, i.e., the preimages of feature vectors exist, the optimal reduction in VC of polynomial classifiers is given by the following theorem,

**Theorem 5.** *For any target polynomial $h^* \in \mathcal{H}$, the optimal teaching set that reduces the VC dimension of the polynomial version space by $B - 1$ is given as $D_T^B = \{(x_i, +1) : \phi(x_i) \perp w^*, i \in [B - 1], \forall i \neq j, \phi(x_i) \perp \phi(x_j)\} \cup \{(\phi(x_B) = -\sum_{i=1}^{B-1} \phi(x_i), +1)\}$.*

We defer the proof to the appendix. Note that the above algorithm relies on computing of preimage feature vectors and we also propose an algorithm to compute them in the appendix.

We recall two major limitations of instance agnostic teaching. First, the teacher does not know $P_\mathcal{X}$ and it is required to be only competitive with respect to instance agnostic solution. However, when teacher knows $P_X$ the optimal teaching set can be much smaller than instance agnostic setting and so a better method is required. Secondly, we can only handle algorithms with specialized hypothesis

classes as the problem of VC reduction is a hard problem in general. This motivates us to consider an instance aware teaching with function approximation where we aim to tackle both the issues.

## 5 Instance Aware Teaching Setting

In an instance aware setting, the teacher knows the environment distribution $P_{\mathcal{X}}$ and aims to find a $B$-budgeted teaching set that minimizes the expected risk at the end of the nature phase.

**Expected risk**: Given an instance aware instance $(\mathcal{X}, \mathcal{Y}, P, h^*, \mathcal{H}, \delta, n, B)$, the expected optimal NtN teaching objective is defined as follows :

$$D_T^* \leftarrow \underset{D_T : |D_T| \leq B}{\arg \min} \ \mathbb{E}_{D_E \sim P^n} \left[ R_P(\mathcal{A}(D_T \cup D_E)) \right].$$

We note that, unlike the previous setting, the objective requires the teacher to produce a teaching set that is competitive w.r.t. the instance specific $P$ which can be much smaller than instance agnostic solution. Also, this setting can handle any learning algorithm $\mathcal{A}$. Computing a close form of the risk of training an algorithm is a challenging problem.

To handle this, we take a function approximation approach by first approximating the risk using a datamodel Ilyas et al. (2022) and then uses this risk model to solve NtN. For simplicity, we focus mainly on linear datamodels but our methodology extends to more complex input differentiable function approximators like deep neural networks. We refer the interested readers to the appendix for a clean outline on extending our method to neural datamodels using projected gradient descent.

**Linear Datamodel**: Linear datamodel proposed by Ilyas et al. (2022) aims to approximate the risk $R(\mathcal{A}(D))$ of training an algorithm $\mathcal{A}$ on a dataset $D$ as a linear function of a dataset.

More formally, given a pool of input universe $\mathcal{X}$, the risk of training an algorithm $\mathcal{A}$ on dataset $D$ is modeled as a linear function in the indicator feature representation of dataset $D$, i.e., $\mathbb{1}_D \in \{0,1\}^{\mathcal{X}}$.

$$R(\mathcal{A}(D)) = w_P^\top \mathbb{1}_D. \tag{7}$$

The parameter $w_P$ can be estimated directly by solving a meta-learning problem on meta-dataset $\mathcal{D} = \left\{ \mathbb{1}_{D_i}, R(\mathcal{A}(D_i)) \right\}|_{i=1}^m$ sampled from a distribution defined over possible data subsets $P_{2^{\mathcal{X}}}$,

$$w_P \leftarrow \arg \min_w \frac{1}{m} \sum_{i=1}^m \ell_2(w^\top \mathbb{1}_{D_i}, R(\mathcal{A}(D_i))) + \lambda \|w\|_1. \tag{8}$$

**Remark 4.** *Note that this method works for any learner $\mathcal{A}$ and hypothesis class $\mathcal{H}$, as long as one can efficiently train the base learner $\mathcal{A}$ on a collection of datasets $\mathcal{D}$.*

Once we have $w_P$, the function $w_P^\top \mathbb{1}_D$ serves as a surrogate for true risk which is then used to solve the original NtN problem under the following realizability assumption.

**Assumption 1** (Realizability of Linear Datamodel)**.** *The risk function of learning algorithm $\mathcal{A}$ is realizable under linear datamodel iff $R(\mathcal{A}(D)) = w_P^\top \mathbb{1}_D, \forall D$.*

**Algorithm using Linear Datamodel**: Using linear datamodel under assumption 1, the risk in 7 can be expressed as $R(\mathcal{A}(D \cup D_T)) = w_P^\top \cdot \mathbb{1}_{D \cup D_T}$, which simplifies NtN objective to:

$$D_T^* \leftarrow \min_{D_T : |D_T| \leq B} w_P^\top \cdot \mathbb{E}_D[\mathbb{1}_{D \cup D_T}]. \tag{9}$$

Expanding $E_D \left[ \mathbb{1}_{D \cup D_T} \right]_x = (1 - (1 - P_x)^n) + \mathbb{1}_{x \in D_T} \cdot (1 - P_x)^n$, we note that the first term is independent of $D_T$ and thus can be ignored. This reduces equation 9 to the following,

$$D_T^* \leftarrow \arg \min_{D_T : |D_T| \leq B} \sum_{x \in \mathcal{X}} \mathbb{1}_{x \in D_T} \cdot w_{P,x} (1 - P_x)^n. \tag{10}$$

This is a *Unit Profit Knapsack problem* where every item $x$ has a unit cost and weight $w_{P,x}(1 - P_x)^n$. It is efficiently solvable by choosing $B$ items with smallest weight leading to the following theorem.

**Theorem 6.** *Under assumption 1, the instance-aware NtN problem is efficiently solvable and the optimal solution is given by,*

$$D_T^* \leftarrow \arg \min_B \{ w_{P,x} (1 - P_x)^n : x \in \mathcal{X} \}. \tag{11}$$

We remark that in contrast to the solution of Engstrom et al. (2024) which can be interpreted as single-phase teaching, our algorithm also utilizes $P$ to be instance-aware in the nature phase.

## 6 EXPERIMENTS

We evaluate our teaching algorithms for teaching learners with diverse hypothesis classes (in both NtN settings) and compare their performance against relevant baselines, as outlined below:

1. **Performance of the algorithm**: 1.) *Teach vs no-teach*: Given a fixed teaching budget, do our teaching algorithms produce significant gains compared to no teaching? 2.) *Role of budget*: Does higher budget lead to better teaching performance?

2. **Comparison with simulated teaching**: In *simulated teaching* (Sim-Teach), the teacher simulates $K$ learners(we choose a moderate $K$ for fair comparison based on computation cost), each with a random teaching set of size $B$ and then select the best-performing teaching set one among them for final teaching.

Our experiments, designed for conceptual clarity, serve as a clear proof of concept to corroborate our theory. A natural extension of our work would involve more complex benchmarks and datamodels. This represents a promising direction for future work, and we outline a path for it in the appendix.

### 6.1 INSTANCE AGNOSTIC TEACHING BY OPTIMAL VC REDUCTION

We apply our optimal VC reduction algorithm(OPT-VC) to a version space learner with a linear and a axis-aligned rectangle class to demonstrate its effectiveness in instance-agnostic setting.

### 6.1.1 HOMOGENEOUS LINEAR CLASSIFIERS

We consider teaching a $w^* \in \mathbb{R}^4$ to a homogeneous linear version space classifier. The nature's $P$ is a uniform distribution over sphere $\mathbb{S}^4$. We discretize the weight space and do exact version space learning, as specified in equation [2]. The error is computed by evaluating the worst classifier in the version space $\mathcal{V}(D)$ on a held-out test set.

**Our results**: We tested our algorithm 4 for teaching this learner on various budget $B \in \{0, \dots, d+1\}$ and plot its NtN performance $\hat{R}(\mathcal{A}(D_T \cup D_E))$ as a function of $n_{iid}$ as shown in Figure 3(a).

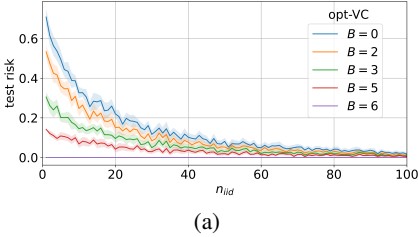

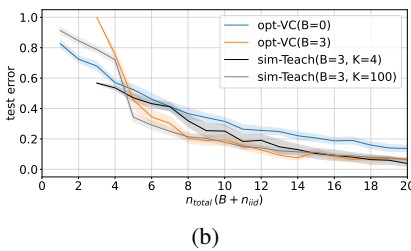

|       |       |
|-------|-------|
| (a)   | (b)   |

Figure 3: (a.) Performance of our OPT-VC algorithm on a linear learner. (b.) Comparing our optimal VC reduction algorithm to simulated baseline for teaching a linear learner.

The blue curve $B = 0$ denotes no-teaching, i.e., the learner only learns from $n_{iid}$ points while other curves represent teaching with respective budget $B$. Figure 3(a), we observe that OPT-VC outperforms no-teaching for all $B$. Moreover, a higher $B$ consistently leads to lower test risk, reaching to zero for all $n_{iid}$ once $B \geq TD = 5$. We also compare our algorithm against the Sim-Teach with $B = 3$ and $K = 4, 100$ simulations as shown in Figure 3(b). We observe that Sim-Teach with $K = 4$ performs a bit better than i.i.d. teaching but is still outperformed by OPT-VC. Even with $K = 100$, which is computationally expensive, Sim-Teach could barely compete with our OPT-VC algorithm.

### 6.1.2 AXIS ALIGNED RECTANGLE CLASS

We consider axis-aligned rectangle class defined on space $\mathcal{X} = \{-n, \cdots, n\}^2$ and choose a target rectangle $h^*$ and $P_{\mathcal{X}} = U(\mathcal{X})$. As before, the verion space learner maintains a version space $\mathcal{V}(D; \mathcal{H})$ and is evaluated by the worst hypothesis in $\mathcal{V}(D)$.

**Our results**: Figure 4(a) shows the NtN performance of our OPT-VC algorithm on various budget sizes 1 as a function of $n_{iid}$ on x-axis.

As before, the blue curve corresponds to no teaching while others represent budgeted teaching with various $B$. We see from Figure 4(a) that our OPT-VC consistently outperforms no-teaching and leads to lower error with higher budget. Again, once $B \geq TD = 6$, nurture alone leads to a zero risk.

We also compare OPT-VC to Sim-Teach and show the results in Figure 4(b). Sim-Teach simulates $K = 4,100$ learners with a random $B = 3$ teaching set and picks the best one it finds. Unlike linear case, Sim-Teach significantly underperforms w.r.t. our OPT-VC algorithm on both $K$'s.

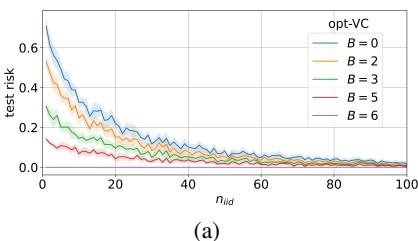 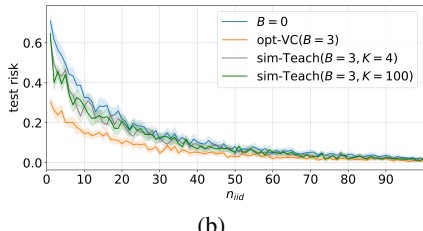

|             (a)             |             (b)             |

Figure 4: (a.) Performance of our optimal VC reduction algorithm on learner, and (b.) Comparing our optimal VC reduction algorithm to the simulated baseline in teaching axis-aligned rectangle class.

## 6.2 INSTANCE AWARE TEACHING THROUGH LINEAR DATAMODEL

In this section, we demonstrate the effectiveness of our linear datamodel algorithm to teach a linear perceptron learner in the instance-aware setting of section [5].

For simplicity, we choose a finite-size universe, $\mathcal{X} \subset \mathbb{R}^2$, consisting of equally-spaced points on the unit circle, a $w^* \in \mathbb{R}^2$ and $P_{\mathcal{X}} = U(\mathcal{X})$ as shown in Figure 5(b). We first train a linear datamodel $w_P$ that represents the risk of linear perceptron(refer to the appendix for more details). Once we obtain $\hat{w}_P$, we select the bottom $B$ input $\mathcal{X}$'s as teaching dataset based on value $\hat{w}_{P,x}(1 - P_x)^n$.

**Our results**: We evaluate the NtN risk of perceptron on the teaching dataset produced by datamodel method(OPT-DM) and report it in Figure 5(a). We observe that OPT-DM with budget $B = 2, 3$ significantly outperforms no-teaching $(B = 0)$. It is also worth noting that $D_T^*$ generated by OPT-DM differs somewhat from those produced by OPT-VC, as illustrated in Figure 5(b). Nevertheless, both approaches reduce the learner's risk compared to just using i.i.d. dataset, as shown in Figure 5(a).

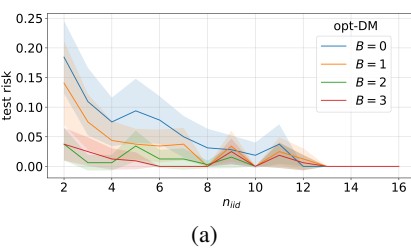 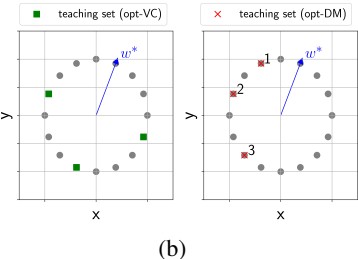

|             (a)             |             (b)             |

Figure 5: (a.) Comparing the teaching set (constructed with linear datamodel) with the case of no teaching. (b.) Teaching sets as constructed by linear datamodel method on a perceptron learner in $\mathbb{R}^2$.

## 7 LIMITATIONS & CONCLUSION

This work advances the study of machine teaching by considering a more realistic two-phase setting where the teacher is constrained by a budget. We proposed novel and efficient algorithms to cater to different assumptions on the teacher's capabilities. We provided theoretical guarantees to our algorithms and demonstrated their effectiveness through experiments against strong baselines.

Looking forward, two promising avenues for research emerge. First, while optimizing the VC dimension using a teaching set is NP-hard in general, the design of approximation algorithms is an important open problem. Second, understanding limitations of linearity and developing non-linear datamodels further could significantly enhance its practical impact on large-scale applications.

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

# A APPENDIX

# B PROOFS

## B.1 FINITE BINARY HYPOTHESIS CLASS

**Budgeted Maximum Coverage Problem**: Given a finite universe of items $U$ and a finite collection of subsets of the universe $V = \{V_x \subseteq U : x \in \mathcal{X}\}$, where $\mathcal{X}$ is a finite set, the goal is to find a subcollection of $V$ of size upto $B$, that covers maximum number of elements of $U$. This problem is known to be NP-hard Khuller et al. (1999). However, a greedy algorithm that greedily chooses a subset to reduce $U$ maximally is approximately optimal, and achieves an approximation ratio of $1 - \frac{1}{e}$. This leads to the following guarantee on the optimal reduction of the version space size.

**Theorem 7** (Theorem 2 of main text). *There exists an algorithm that reduces the version space size of finite hypothesis class up to an approximation ratio of $1 - \frac{1}{e}$.*

*Proof.* We note that each demonstration $(x, h^*(x))$ eliminates a subset of hypothesis, $V_x = \{h \in \mathcal{H} : h(x) \neq h^*(s)\}$ from $\mathcal{H}$. Maximally reducing the size of the version space requires eliminating as many hypotheses from $\mathcal{H} \backslash \{h^*\}$ as possible under budget $B$. This is nothing but budgeted maximum coverage problem with $U = \mathcal{H} \backslash \{h^*\}$, $\{V_x : x \in \mathcal{X}\}$ as defined above and the result follows from Khuller et al. (1999). □

## B.2 AXIS-ALIGNED RECTANGLES ON $\mathbb{Z}^2-$GRID

**Definition 2** (Extending $h$ to $h'$). *A rectangle $h'$ is said to extend $h$ if $\{x : h(x) = 1\} \subsetneq \{x : h'(x) = 1\}$. An extension can occur along one or more of the four sides[1] of the rectangle — namely, top, bottom, left, or right.*

**Definition 3** (Fixing sides and degrees of freedom). *Given a version space $\mathcal{V}$ that contains $h^*$, we define the **degrees of freedom** of $\mathcal{V}$ w.r.t. $h^*$ as the number of sides along which $h^*$ can be **extended** to another rectangle $h'$, such that $h' \in \mathcal{V}$. If no such extension is possible along a particular side, we say that that side is **fixed** in $\mathcal{V}$.*

**Remark 5** (Reducing degrees of freedom). *The original hypothesis class has four degrees of freedom corresponding to the four sides along which $h^*$ can be independently extended while still remaining within the version space $\mathcal{V}$. When $k \in \{1, 2, 3, 4\}$ sides of $h^*$ are **fixed** in $\mathcal{V}$, the degrees of freedom of the version space reduce by $k$.*

**Lemma 8.** *Let $\mathcal{H}$ be the class of axis-aligned rectangles on $\mathbb{Z}^2$-grid. For any rectangle, $h \in \mathcal{H}$, fixing one (two) of its sides requires two (three) labelled examples.*

*Proof. (Fixing one side).* Without loss of generality, consider **fixing** $y_{\min}^*$, corresponding to the bottom side of the target rectangle $h^*$. This can be done using exactly two labelled examples: $\{((x, y_{\min}^*), +), ((x, y_{\min}^* - 1), -)\}$ where $x_{\min} \leq x \leq x_{\max}$. The '+' and '−' examples force every consistent hypothesis $h$ to satisfy $y_{\min} \leq y_{\min}^*$ and $y_{\min} > y_{\min}^* - 1$, respectively, thereby enforcing $y_{\min} = y_{\min}^*$. Thus, no **extensions** are possible along the bottom side thereby **fixing** this side. With only one labelled example there is always an **extension** $h$ of $h^*$ possible along the bottom side such that $h$ is consistent with the labelled example — enlarge downward (shrink upward) given a single positive (negative) example. Hence two examples are necessary.

*(Fixing two sides).* Naively, by the reasoning above, we can use four examples to **fix** two sides. But we can do better by using just three examples: labeling a corner point of the rectangle as '+' and two adjacent points just outside the rectangle as '−'. For e.g., if the corner is $(x_{\max}, y_{\min})$, then the following set suffices as a teaching set: $\{((x_{\max}, y_{\min}), +), ((x_{\max} + 1, y_{\min}), -), ((x_{\max}, y_{\min} - 1), -)\}$. The necessity of three labelled examples is apparent given the need for two labelled examples to **fix** a single side (as seen above). □

**Theorem 9** (Theorem 3 of main text). *The VC dimension of axis-aligned rectangles in $\mathbb{Z}^2$ can be optimally reduced as shown in the Table 9 below:*

---

[1]By a side of a rectangle, we mean one of the 4-tuple values that defines any rectangle $h = \{x_{\min}, x_{\max}, y_{\min}, y_{\max}\} \in \mathcal{H}$. For e.g., $x_{\min}$ refers to the bottom-side of the rectangle $h \in \mathcal{H}$. We follow this convention in the subsection B.2 for readability of the proofs.

| Budget $B$ | 1 | 2 | 3 | 4 | 5 | $\geq 6$ |
|------------|---|---|---|---|---|----------|
| min VC | 4 | 3 | 2 | 2 | 1 | 0 |

Table 2: Minimum VC achievable by $B$-budgeted teaching on axis-aligned rectangle class in $\mathbb{Z}^2$.

*Proof.* We will proceed by starting with the case of $B = 2$ and end with the case $B = 5$ in that increasing order. The case $B \geq 6$ follows from classical Teaching Dimension Goldman & Kearns (1995) as $TD = 6$.

*Case $B = 1$:* We cannot **fix** any side of the target $h^*$ with $B = 1$ (Lemma 8) and, hence, VC-dimension remains 4.

*Case $B = 2$:* We can **fix** exactly one of the sides of the target $h^*$ with two examples as per Lemma 8. This means the reduced version space $\mathcal{H}'$ has 3 degrees of freedom (Remark 5) and $VC(\mathcal{H}') = 3$: Consider the rightmost side (i.e. $x_{\max}$) is **fixed** and take four points in general position. If one point lies within the convex hull of the other three, fixing $x_{\max}$ prevents labeling the outer three points as '+' and the interior point as '−'. Otherwise, if no point is inside the convex hull of the remaining three points, label the two points farthest apart (along the axis of the **fixed** side) as '+' and the remaining two as '−'. In both cases, at least one labeling is impossible, implying $VC(\mathcal{H}') < 4$. However, the set $\{(x, y), (x + 1, y + 1), (x + 1, y − 1)\}$, where $x < x_{\max}$, can be shattered.

*Case $B \in \{3, 4\}$:* **Fix** two opposite sides (e.g., $x_{\min}$ and $x_{\max}$) via four examples (Lemma 8). Thus, the reduced version space $\mathcal{H}'$ has 2 degrees of freedom (Remark 5) and $VC(\mathcal{H}') = 2$: Any three collinear points lying between these sides cannot all be labelled arbitrarily (one of them becomes the 'middle' point). Alternatively, for a triplet in general position, if one of the points falls outside these sides, we cannot flip its label without contradiction. Hence no triple is shattered, but pairs are.

Alternatively, by Lemma 8, use three labelled examples to **fix** two sides that meet at a corner of the target rectangle (e.g., $(x_{\max}, y_{\min})$). Again, we have reduced version space $\mathcal{H}'$ with 2 degrees of freedom (Remark 5). This reduction ensures no set of three points can be shattered: if they are collinear, the 'middle' point cannot be labelled differently from the other two; if they are in general position, at least one labelling is impossible (e.g. two points that are closest to one of the sides to be **fixed** are labelled as '+' and the remaining point as '−'). However, two points remain shatterable (for instance, by choosing a suitable $(x, y)$ for the top-right corner). Hence, $VC(\mathcal{H}') = 2$.

*Case $B = 5$:* With three sides **fixed** by, for instance, **fixing** a corner and one of the sides corresponding to the opposite corner (Lemma 8) using three and two labelled examples, respectively, the version space reduces to $\mathcal{H}'$ with 1 degree of freedom (Remark 5) and $VC(\mathcal{H}') = 1$: No two-point set can be shattered as one labeling always becomes impossible depending on which three of the 4-tuple values have been taught. However, we can construct a single point set that can be labeled in any way.

*Case $B = 6$:* Since $TD(h^*; \mathcal{H}) = 6$, we can simply use a teaching set of size six so that the version space is reduced from $\mathcal{H}$ to $\mathcal{H}' = \{h^*\}$. Thus, $VC(\mathcal{H}') = 0$. $\qquad\square$

### B.3 HOMOGENOUS LINEAR CLASSIFIERS

Let $D$ be a dataset of size $m$ with $p$ negative labels generated by a target hypothesis $w^* \in \mathcal{H}_{\text{linear}}$, given as follows,

$$D := D^- \cup D^+ := \{(x_i, -1) : i \in [p]\} \cup \{(x_i, +1) : i \in \{p + 1, \cdots, m\}\}. \qquad (12)$$

Note that $D$ induces a polyhedral cone as a version space,

$$\mathcal{V}(D) = \{w : w^\top x_i < 0, \forall x_i \in D^-, w^t x_i \geq 0, \forall x_i \in D^+\}.$$

Since a cone lies in the subspace span of its vectors, we have that $VC(\mathcal{V}(D)) \leq d(\mathcal{V}(D))$, where $d$ denotes dimensionality. Next lemma shows that $VC(\mathcal{V}(D))$ is also lower bounded by $d(\mathcal{V}(D)) - 1$.

**Lemma 10.** *For a dataset $D$ containing all positive labels, i.e., $D = D^+$, we have that $VC(\mathcal{V}(D)) \geq d(\mathcal{V}(D))$. Otherwise, $VC(\mathcal{V}(D)) \geq d(\mathcal{V}(D)) - 1$.*

*Proof.* Let $\mathcal{V}(S) = \{w \in \mathbb{R}^D : w^\top x_i \geq 0, \forall i \in [n]\}$ be a closed polyhedral cone formed by an all positive dataset $S$ and let $l$ be its dimensionality. We show that $VC(S) = l$.

We construct a set $V$ consisting of $l$ points in $\mathbb{R}^d$ and show that it can be shattered by $S$. For each labeling $s \in \{0,1\}^l$, we construct a labeling vector $w_s \in S$ that achieves label $s$ on $V$. Consider a set of $l$ orthogonal vectors in $S$ and arrange them as columns of matrix $A = [v_1, v_2, \cdots, v_l] \in \mathbb{R}^{d \times l}$. Since, $S$ is $l$-dimensional, we can always find such a set of vectors.

Now, we show that the set of $l$ points, $V = \{-A^{-\top}e_1, -A^{-\top}e_2, \cdots, -A^{-\top}e_l\}$ can be shattered by $S$. We use pseudo-inverse if $l < d$.

Let $s \in \{0,1\}^l$ be a labeling vector. We will show that $w_s = \left(\sum_{i:s_i=0} Ae_i\right)$ achieves the labeling $s$ on $V$. First, note that since $S$ is a convex cone, $w_s = \sum_{i:s_i=0} v_i \in S$.

We have that, $(-A^{-\top}e_j)^\top w_s = \sum_{i:s_i=0} -e_j^\top A^{-1} Ae_i = \sum_{i:s_i=0} -e_j^\top e_i = -\mathbb{1}[s_j = 0]$. Thus, $h_{w_s}(-A^{-\top}e_j) = \mathbb{1}[(A^{-\top}e_j)^\top w_s \geq 0] = \mathbb{1}[s_j = 1]$ and so $w_s$ realizes the labeling $s$. Since, $dim(S) = l, S \subseteq \mathbb{R}^l$, and we have that $VC(S) \leq VC(\mathbb{R}^l) = l$. Thus, $VC(S) = l$.

Now, if $S$ has an open halfspace, i.e., the corresponding dataset contains a negative labeled point, then all the labeling except all positive one can be realized, i.e., $w_{\not\Vdash} = 0$ does not lie in $S$, while rest all $w_s$ still lie in $S$ and above proof proceeds. Thus, $l - 1 \leq VC(S) \leq l$. $\square$

Next, we characterize the dimensionality of the version space and asserts that maximal dimensionality reduction can be achieved by $B$ positively labeled demonstrations as stated in point 3 below.

**Lemma 11.** *The following statement hold true for a consistent dataset $D$ generated by $w^* \in \mathbb{R}^D$ :*

1. *Negative points do not help in reducing dimensionality, i.e., $d(\mathcal{V}(D)) = d(\mathcal{V}(D^+))$.*

2. *$\forall D : |D| \leq B$, we have that $d(\mathcal{V}(D)) \geq d - |D| + 1$.*

3. *$\forall B \leq d$, the dataset $D_T^B$ achieves optimal reduction in VC by $B - 1$, thus, $d(\mathcal{V}(D_T^B)) = d - B + 1$.*

$$D_T^B = \{(v_1, +1), \ldots, (v_{B-1}, +1), (-\sum_{i \in [B-1]} v_i, +1)\}$$

   *where $\{v_1, \ldots, v_{B-1}\}$ is a $B$-basis of $w^{*\perp}$ subspace.*

*Proof.* To prove 1, we start with a basis set of $\mathcal{V}(D^+)$ and a feasible $x_0 \in \mathcal{V}(D)$. Translating the basis set by $x_0$ yields a basis set for $\mathcal{V}(D)$. To prove 2, we can construct a dataset with $|D|$ points that kills $|D| - 1$ vectors in orthogonal subspace of $w^*$. For prove 3, it is easy to see that a classifier $w$ is consistent on $D_T^B$ if and only if $w \notin \text{span}(v_1, \cdots, v_{B-1})$. Thus, the dimensionality of the version space $\mathcal{V}(D_T^B)$ reduces from $d(\mathbb{R}^d) = d$ to $d(\mathcal{V}(D_T^B)) = d - B + 1$. $\square$

**Theorem 12** (Theorem 4 of main text). *There exists an algorithm that $\forall B \leq d + 1$, optimally reduces the VC of the linear class to $d - B + 1$ and the optimal teaching set is given as $D_T^B = \{(v_1, +1), \ldots, (v_{B-1}, +1), (-\sum_{i \in [B-1]} v_i, +1)\}$, where, $\{v_1, \ldots, v_{B-1}\}$ is a $B$-basis of $w^{*\perp}$ subspace.*

*Proof.* We make the following observations:

1. For a dataset with all positive demonstrations, we have that $VC(\mathcal{V}(D)) \geq d(\mathcal{V}(D)) \geq d - B + 1$ and the lower bound is achieved by $D_T^B$ is Lemma 11.

2. For a dataset with at least one negative demonstration, $VC(\mathcal{V}(D)) \geq d(\mathcal{V}(D)) - 1 = d(\mathcal{V}(D^+)) - 1 \geq d - |D^+| \geq d - B + 1$. The first inequality follows from Lemma 10 while others follow from Lemma 11.

Thus, for $B \leq d$, the optimal strategy is to teach dataset $D_T^B$ to kill of $B - 1$ dimensional subspace. We refer to Figure [2] for an illustrative teaching example in $w^* \in \mathbb{R}^3$ and $B = 2, 3$. For $B \geq d + 1$, our optimal teaching dataset matches with the unconstrained teaching dataset proposed by Kumar et al. (2021). $\square$

### B.4 POLYNOMIAL HYPOTHESIS CLASS

**Theorem 13** (Theorem 5 of main text). *For any target polynomial $h^* \in \mathcal{H}$, the optimal teaching set that reduces the VC dimension of the polynomial version space by $B - 1$ is given as $D_T^B = \{(x_i, +1) : \phi(x_i) \perp w^*, i \in [B-1], \forall i \neq j, \phi(x_i) \perp \phi(x_j)\} \cup \{(\phi(x_B) = -\sum_{i=1}^{B-1} \phi(x_i), +1)\}$*

*Proof.* The teacher can computes $B - 1$ orthonormal bases functions to $\theta^*$ and their negative summand represented by vectors $v_1, \cdots, v_{B-1}, -\sum_{i \in [B-1]} v_i$ in the standard bases. However, to be a valid teaching set these vectors must be induced by the feature function $\phi$ on certain input set (preimages under $\phi$) $\{x_i \in \mathbb{R}^d : i \leq [B]\}$. Assuming that there exists a set of such inputs, the teacher can construct the optimal teaching set given by the dataset $D_B^T = \{(x_i, 1) : i \in [B-1]\} \cup \{(-\sum_{i \in [B-1]} x_i, 1)\}$. $\square$

**Computing the preimages for teaching vectors**: We propose an iterative algorithm that can compute the orthogonal preimages efficiently assuming they exist. At the start of iteration $k \in [B]$, say we have already computed the $k - 1$ orthogonal bases vectors in $w^{*\perp}$, the optimization problem to find the next orthobasis vector $v_k$ is as follows:

$$x_k, v_k \leftarrow \min_{x,v} \quad \|v - \phi(x)\|^2$$

$$\text{s.t. } \alpha^\top v = 0, \forall \alpha \in \{v_i : i \leq k-1\} \cup \{w^*\}$$

Once all $\{v_i\}|_{i \leq B-1}$ vectors have been computed, we compute the preimage of their negative summand by solving

$$x_B, v_B \leftarrow \min_{x,v} \|v - (-\sum_{j \leq B-1} \phi(x_j))\|^2 + \lambda \|v - \phi(x)\|^2.$$

**Remark 6.** *This method extends to any finite dimension kernel. However, not all kernel mapping may admit a pre-image set thereby limiting this approach.*

## C EXPERIMENTS FOR INSTANCE-AWARE TEACHING USING DATAMODELS

**Overview:** We first train a datamodel using meta dataset $D_M := (D_i, R(\mathcal{A}(D_i)))$ computed by training perceptron algorithms on various data subset $D_i$ sampled from $P_D \in \Delta(2^{\mathcal{X}})$ and tested on a held out test set to get the meta label $R(\mathcal{A}(D_i))$. We use a uniform distribution $P_D$ over all subset of size $\leq \alpha \cdot |\mathcal{X}|$ where $\mathcal{X}$ is a finite set of points in $\mathbb{R}^2$.

Once, we have the meta dataset, we train the datamodel parameter $\hat{w}_P$ using sparse linear regression on $D_M$. We then compute the optimal teaching set as $B$ points in $\mathcal{X}$ with minimum weights $\hat{w}_{P,x}(1 - P_x)^n$.

We then evaluate the NtN performance on this dataset for various values of $n_{iid}$. The teaching code for VC reduction teaching for linear and axis aligned rectangles can also be found in the supplementary materials (data_models.ipynb).

Now we describe our setup and pseudo-code along with hyperparameter configurations for estimating linear datamodels, computing the teaching sets and evaluating NtN performance.

### C.1 SETUP, DATA GENERATION AND EVALUATION

We consider teaching a homogeneous linear classifier in $\mathbb{R}^2$. The learner is trained via an ERM procedure using a *perceptron loss* (as a computationally convenient surrogate to 0–1 loss) on a finite universe $\mathcal{X} \subset \mathbb{R}^2$.

We let $\mathcal{X}$ be a set of 16 uniformly-spaced points on the unit circle in $\mathbb{R}^2$. Each point $x \in \mathcal{X}$ is labeled via a target linear separator $w_{\text{true}}$[2]. In the *nature phase*, the learner receives $n$ i.i.d. draws from the uniform distribution $P_{\mathcal{X}}$. Since we have access to all of $\mathcal{X}$ and we know that $P_{\mathcal{X}}$ is uniform, we evaluate the test performance of a linear classifier as the average 0-1 loss on the entire feature space $\mathcal{X}$.

---

[2] For simplicity, we have chosen $w_{\text{true}}$ to be one of these 16 points in our simulations.

## C.2 Teaching through Linear Datamodels (OPT-DM)

As discussed in Section 5, we use linear datamodels to approximate the learner's risk as a linear function of the dataset. We then use these estimated datamodel's weights to find the budgeted teaching set which is to be used for teaching the learner under the prescribed teaching budget.

### C.2.1 Estimating Linear Datamodels

- **Meta-Dataset Construction:** We sample $N_{\text{subsets}}$ subsets $S_i \subset \mathcal{X}$ of size $\alpha \cdot |\mathcal{X}|$ via a distribution $P_{\text{subsets}}$. For each subset $S_i$, we train a *perceptron* on $S_i$ using the perceptron loss and measure its 0–1 test loss $y_i$ on the entire space $\mathcal{X}$. We use $\alpha = 0.25$ and $P_{\text{subsets}}$ to be a uniform distribution in our experiments.

- **Sparse Linear Fit ($\ell_1$-regularization):** We collect pairs $(\mathbf{1}_{S_i}, y_i)$ and solve

$$\theta = \arg\min_{\theta'} \frac{1}{N_{\text{subsets}}} \sum_{i=1}^{N_{\text{subsets}}} \left(\theta'^{\top} \mathbf{1}_{S_i} - y_i\right)^2 + \lambda \|\theta'\|_1.$$

We use `scikit-learn`'s `LassoLarsCV` solver with 4-fold cross-validation to automatically select $\lambda$ and perform the $\ell_1$-regression. Here, $\theta_x$ measures how strongly point $x \in \mathcal{X}$ influences the overall risk. The pseudo-code for estimating datamodels is outlined in Algorithm 1.

---

**Algorithm 1** EstimateDataModel

---

**Require:** Universe $\mathcal{X} \subseteq \mathbb{R}^3$, subsampling fraction $\alpha \in (0,1)$, test-set size $m$, number of subsets $N_{subsets}$, distribution over subsets $P_{subsets}$
1: $T \leftarrow []$      ▷ Initialize datamodel training set
2: $S = \{(x,y) \mid x \in \mathcal{X}, y = 2 \cdot \text{sign}(w_{true} \cdot x) - 1\}$
3: **for** $i = 1$ to $N_{subsets}$ **do**
4:      Sample subset $S_i \subset S$ as per $P_{subsets}$ with $|S_i| = \alpha \cdot d$
5:      Train $\mathcal{A}$ on $S_i$
6:      Sample $D_{test} \sim P_{\mathcal{X}}^m$
7:      $y_i \leftarrow \frac{1}{m} \sum_{(x,y) \in D_{test}} \ell_{0-1}(\mathcal{A}(x; S_i), y)$
8:      Define $\mathbf{1}_{S_i} \in \{0,1\}^d$ where $(\mathbf{1}_{S_i})_j = 1$ if $x_j \in S_i$ else 0
9:      $T \leftarrow T \cup \{(\mathbf{1}_{S_i}, y_i)\}$
10: **end for**
11: $\theta \leftarrow \text{RunRegression}(T)$
12: return $\theta$

---

### C.2.2 Computing the teaching set

Having estimated the linear datamodel using Algorithm 1 above, we use its weights $\theta$ along with the given teaching budget $B$, a nature budget $n$, and underlying data distribution $P_{\mathcal{X}}$, we can compute the limited-budget teaching set by finding $B$ points that minimize the following

$$\sum_{x \in D} \theta_x \left(1 - P_{\mathcal{X},x}\right)^n.$$

We thus pick the $B$ smallest values of $\theta_x(1 - P_{\mathcal{X},x})^n$ as proved in Theorem 6. This is outlined below as Algorithm 2.

---

**Algorithm 2** ComputeTeachingSet

---

**Require:** Teaching budget $B$, weight vector $\theta$, distribution $P_{\mathcal{X}}$, nature budget $n$
1: $D_T \leftarrow \arg\min_B \{\theta_x \cdot (1 - (P_{\mathcal{X},x})^n)\}$
2: return $D_T$

---

### C.2.3 NtN Evaluation

Given that we have estimated the datamodel and computed the limited-budget teaching set $D_T$, we now measure performance $(R_{NtN}(D_T, n_{iid}))$ of the learnt classifier as a function of the nature budget $n_{iid}$. In particular, by sampling $K = 50$ distinct i.i.d. subsets of size $n_{iid}$, training a perceptron on $D_T \cup D_k^n$ for each $k \in [K]$, and computing average test-set error $R_{NtN}(D_T, n_{iid})$ as averaging their 0–1 loss on the entire space $\mathcal{X}$ (Equation equation 13 below). We repeat this for nature budgets $n_{iid} = 1, \ldots, N$ for $N = 16$. Note that $K = 50$ subsets are sampled for each $n_{iid}$ so that we can compute the 95% confidence intervals (shaded regions) as seen in Figure 5(a).

$$R_{NtN}(D_T, n_{iid}) = \frac{1}{K} \sum_{k=1}^{K} \frac{1}{|\mathcal{X}|} \sum_{x \in \mathcal{X}} \ell_{0-1}(\mathcal{A}_k^{n_{iid}}(x), y_x) \qquad (13)$$

where $\{\mathcal{A}_k^n\}$ denotes trained models using the $k^{th}$ subset of i.i.d. training set with $n_{iid}$ nature budget and $y_x$ denotes the true label of any point $x \in \mathcal{X}$ as per $w_{\text{true}}$.

### C.2.4 Combining everything: Training a linear classifier through NtN

The `OPT-DM` procedure as outlined above in B.2.1 – B.2.3 can be collectively expressed as Algorithm 3 below.

---

**Algorithm 3** TrackNtNPerformance

---

**Require:** $\mathcal{X} \subset \mathbb{R}^2$, $|\mathcal{X}| = d$, uniform $P_{\mathcal{X}}$, teaching budget $B$, max nature budget $N$, $\alpha$, $N_{subsets}$, $P_{subsets}$, test-set size $m$, number of models $K$
1: $\theta \leftarrow$ EstimateDataModel($\mathcal{X}, \alpha, m, N_{subsets}, P_{subsets}$)
2: **for** $n_{iid} = 1$ to $N$ **do**
3:     $D_T \leftarrow$ ComputeTeachingSet($B, \theta, P_{\mathcal{X}}, n$)
4:     **for** $k = 1$ to $K$ **do**
5:         Sample $D_k^{n_{iid}} \sim P_{\mathcal{X}}^{n_{iid}}$
6:         Train $\mathcal{A}_k^{n_{iid}} = \mathcal{A}(D_T \cup D_k^{n_{iid}})$
7:     **end for**
8:     Evaluate $R_{NtN}(D_T, n_{iid})$ as per Equation equation 13
9: **end for**

---

## C.3 Extending to Neural Datamodel in Instance-Aware Setting

Our Instance-Aware method based on datamodel is very generic and in fact it can be extended to any datamodel that can be optimized over input space and that includes deep neural networks as well.

We briefly outline the procedure for using neural datamodel for NtN teaching. For teaching purpose, we assume that we already have an access to neural datamodel (one can easily train one similar to how we trained linear datamodel using same underlying metadataset just by substituting linear function class with neural network function class).

$$\theta^* \leftarrow \arg\min_{\theta} \frac{1}{m} \sum_{i=1}^{m} \ell_2(f_\theta(\mathbb{1}_{D_i}), R(\mathcal{A}(D_i))) + \lambda\|\theta\|_2^2.$$

Once we have a trained neural datamodels parameterized by $\theta^*$, we use the following projected gradient descent algorithm to find the best NtN teaching set under budget constraints.

We tested this method on a simple threshold classification problem with input space $\mathcal{X} = \{-4, -3, -2, -1, 0, 1, 2, 3\}$. The ground truth classifier is $h(x) = \mathbb{1}[x \geq 0]$.

We apply our algorithm ProGrad-Ntn using Adam optimizer with a regularization coefficient of 0.1 and learning rate of 0.01 for 10K iterations or until iterates converge to a local minima. The resulting $x^*$ so obtained is $x^* = [-0., 0.01, 0.37, 0.23, 0., 0.01, 0.]$. Projecting on $l_0(x^*) = 2$, yields the dataset $x = [-0.33, 0]$ as a teaching set which indeed is an optimal teaching set for the problem.

---

**Algorithm 4** ProGrad-NtN : Projected Gradient for Nurture-then-Nature

---

**Require:** Model function $f_\theta : x \to \mathbb{R}$, Target $y^* = 0$, Budget $B$, Learning rate $\eta$
**Ensure:** Optimized dataset $D^* \in \{0, 1\}^d$ with $|D^*|_0 \leq B$.
 1: Initialize $D \in \{0, 1\}^d$ randomly.
 2: **while** not converged **do**
 3:    Compute gradient: $\nabla \leftarrow \nabla_D \ell(f_\theta(D), y^*)$
 4:    Perform gradient step: $D \leftarrow D - \eta\nabla$
 5:    Project $D$ onto $\ell_0 \leq B$ ball: Keep the top-$B$ entries of $D$ and set others to 0.
 6: **end while**
 7: **return** $D$

---

We would like to emphasize that the aim of the this and main experiments in the paper serves as a empirical proof of concept for usefulness of linear and neural datamodel. A complete treatment of these methods on complex problems is beyond the scope of this paper and we hope that future works could build on our work to solve more real world NtN problems using our algorithm.

## D   COMPUTE SYSTEM AND LLM USAGE

All code has been run on $48$ core Intel Xeon system with 192GB memory and Macbook M1 Pro laptop with 16GB memory. We have only used LLMs for rephrasing certain points in the paper.

