# OpenReview forum: "Optimal Dataset Design for Nurture-then-Nature Teaching"
_ICLR.cc/2026/Conference — Submitted to ICLR 2026_

### Official Review · Reviewer_G9gh · 2025-10-25

**Soundness:** 3
**Presentation:** 2
**Contribution:** 2
**Rating:** 4
**Confidence:** 3

**Summary:**

This paper studies the machine teaching problem under a two phase framework, where the student first learns from a teacher (nurture), then without the teacher’s supervision on i.i.d. data (nature). The teacher is constrained to a limited dataset size with the goal os minimizing the final error of the student. The authors provide theoretical results for this framework and experimental setups to support these results.

**Strengths:**

The setup of the problem is interesting and similar to real world examples, where the teacher trains the student during the nurture phase e.g., pre-training to get a good initialization so that the student can effectively continue learning without the teacher's supervision after.

**Weaknesses:**

- Practicality of the Instance-Aware Setup. he instance-aware teaching framework assumes that the teacher has knowledge of all examples observed by the student during the nature phase which seems like a strong assumption.
- Clarity and Messaging in Figures. The figures are useful for conveying insights, but some are challenging to interpret. For example, in Fig. 2 and 5b, the key differences or takeaways are not immediately clear (the plots in Fig. 2 appear quite similar). Adding clearer annotations or captions could help readers more easily grasp the intended message. Fig 5b, shows that there are different points selected but it does not help with understanding why these particular points are optimal in each case.
- Further insights from the selected dataset. It could be interesting to visualize (e.g., with a toy vision dataset) the difference in data selected by the teacher in the nurture phase under the ntn setting vs a nature-only setting, under the same budget. Also how the selected examples change under more extreme budget constraints.

**Questions:**

- What are some scenarios where the teacher would be aware of what the student will see in the Nature phase?
- Given than the paper focuses on B < TD, do the selected points reflect something like importance or influence? How does the selected points change with lower values of B.

---

> ### Author Response · Authors · 2025-11-24
>
> We thank the reviewer for the positive assessment of our problem setup and for recognizing the real-world relevance of the "Nurture-then-Nature" framework. We appreciate the constructive feedback on the figures and practical interpretations.
>
> ## 1. When is the teacher aware of the student sample in the Nature phase?
>
> We would like to clarify that "Instance-Awareness" does not imply the teacher knows the exact random sequence of future samples. Instead, it assumes that the teacher knows the underlying distribution $P$ of the nature phase. This is realistic in many scenarios (e.g., Domain Adaptation or Transductive Learning) where the teacher has access to the unlabeled target pool or can sample enough examples to approximate $P$. Alternatively, we have also studied the instance-agnostic setting where the teacher does not need to know $P$ and provided theoretical guarantees for that setting.
>
> ## 2. Effect of Budget $B$ and Figure Clarity (Fig. 2 & 5b)
>
> We thank you for this question. Figure 2 directly illustrates the impact of lower budgets for a linear learner: with $B=3$, the teacher selects points to eliminate a 2-dimensional orthogonal subspace to $w^*$. Reducing the budget to $B=2$ allows the teacher to eliminate only one dimension (point $w_1$), leaving a larger version space and requiring more effort from the learner in the nature phase. This aligns with our theory in Section 4.3, which characterizes optimal teaching as progressively "killing" orthogonal directions. Regarding Figure 5b, it contrasts the geometrically precise set from OPT-VC (killing the subspace) with the set found by OPT-DM. While the OPT-DM points lack simple geometric intuition due to the linear approximation, Figure 5a confirms their effectiveness in minimizing risk compared to no teaching ($B=0$).
>
> ## 3. Visualizing Teacher vs. Nature-only Selection
>
> We would like to clarify that in the Nature-only setting, there is strictly no teaching selection happening. By definition, the Nature phase consists of the learner receiving i.i.d. samples from the environment without any teacher intervention. Therefore, a visualization comparing the data "selected" by the teacher vs. Nature is not applicable, as Nature does not select  - it samples based on density.
>
> We hope these clarifications address your concerns and would be happy answer any follow up question that you may have.

---

### Official Review · Reviewer_PaNq · 2025-10-31

**Soundness:** 3
**Presentation:** 3
**Contribution:** 3
**Rating:** 6
**Confidence:** 2

**Summary:**

This paper studies “Nurture-then-Nature (NtN)” dataset design: a teacher first provides a small, curated labeled set (“nurture”), after which the learner continues on naturally sampled data (“nature”). The work analyzes two regimes: Instance-agnostic teacher: doesn’t know the future data distribution and aims to pick examples that shrink the remaining hypothesis space before the nature phase. and Instance-aware teacher: assumes a linear datamodel where final risk is exactly a linear function of example indicators

**Strengths:**

1. The paper is well written and motivated, and lays out the two-phase training and splits the nature stage into agnostic vs. aware cases.

**Weaknesses:**

1. The “optimality” proof for the agnostic teacher only optimizes an upper bound on risk (via VC dimension). The authors equate minimizing this bound with minimizing true post-nature error, which is not guaranteed.

2. The intro section could benefit from a comparison against curriculum learning, active learning, or modern data selection methods.

3. minor: the paper still shows the template title

**Questions:**

1. Could the authors discuss potential pathways for extending these ideas to larger-scale deep learning settings?

---

> ### Author Response · Authors · 2025-11-24
>
> We thank the reviewer for the positive assessment, for finding the paper "well written and motivated," and for appreciating the clear separation between the agnostic and aware regimes. We address your specific technical questions below:
>
> ## 1. Optimizing the Upper Bound (VC Dimension) vs. True Risk
>
> You correctly point out that in the Instance-Agnostic setting, we optimize the VC dimension (Eq. 5), which minimizes an upper bound on risk. We emphasize that this is the **mathematically optimal minimax strategy** for a teacher facing an unknown environment. Since the teacher is "blind" to the future distribution $P_X$ (Equation 3), minimizing the true expected risk directly is impossible. Instead, the teacher must minimize the worst-case error bound across _all_ possible distributions. As established in **Theorem 1**, minimizing the VC dimension of the surviving version space is the necessary and sufficient condition to satisfy this objective, ensuring robustness against any distribution Nature might present.
>
> ## 2. Deep Learning Extensions & Scope
>
> We outline two concrete pathways to extend our framework to deep learning:
>
> - **Kernel Methods (NTK):** Leveraging the correspondence between infinite-width networks and kernels, our results on Kernel classifiers (**Theorem 5**) offer a theoretical mechanism for teaching Neural Tangent Kernels via feature space preimages.
>
> - **Neural Datamodels:** For non-linear learners, we relax the linear assumption in **Appendix C.3** by employing a neural network $f_\theta(D)$ as the datamodel. We provide **Algorithm 4 (ProGrad-NtN)**, which uses projected gradient descent to optimize teaching sets, effectively creating an NtN-specific "Data Distillation" pipeline.
>
> - **Foundational Scope:** As this is the **first paper** to formalize the NtN setting, our primary goal was to establish theoretical solvability and hardness results. Just like any pioneering work in a new direction, there are natural extensions - such as scaling these algorithms to complex Deep Learning benchmarks - that we identify as promising avenues for future research to address, building on the algorithmic foundations laid here.
>
>
> ## 3. Comparison to Curriculum Learning (CL) and Active Learning (AL)
>
> We will update the Related Work (Section 2) to explicitly contrast NtN with these fields:
> - **vs. Active Learning:** AL is learner drive where learner queries teacher on most important points, whereas NtN is teacher driven - where the teacher proactively constructs the dataset before learning begins.
> - **vs. Curriculum Learning:** Traditional notions of CL is akin to unconstrained machine teaching where there is no budget constraints on the teacher. NtN is defined by a hard budget constraint $B < TD$ (Remark 2.1), forcing the teacher to argue about learner future nature performance rather than just finding a minimal dataset to exactly identify the target hypothesis.
>
>
> ## 4. Title Correction:
> We apologize for the template title oversight and will correct it in the revised manuscript.
>
> We hope these clarifications address your concerns. We are happy to answer any further questions during the discussion period.

---

### Official Review · Reviewer_RAFk · 2025-11-01

**Soundness:** 3
**Presentation:** 3
**Contribution:** 3
**Rating:** 2
**Confidence:** 5

**Summary:**

This paper introduces the Nurture then Nature (NtN) teaching framework: a teacher with limited budget B first provides a teaching dataset (Nurture) and then the learner receives i.i.d. data from the environment (Nature). The teacher’s objective is to minimize the learner’s final error after the Nature phase.

•	Two teacher knowledge regimes are studied:
o	Instance agnostic: teacher does not know PX. The problem is reduced (via PAC guarantees) to minimizing the VC dimension (or other proxies) of the surviving version space; algorithms/constructive solutions or approximations are given for several hypothesis classes (finite binary classes → greedy 1−1/e approximation via budgeted max coverage; axis aligned rectangles → exact VC reductions per budget; homogeneous linear classifiers → kill orthogonal subspace to reduce VC from d to d−B+1; polynomial kernels → analogous feature space construction assuming preimages).
o	Instance aware: teacher knows PX. Using a linear datamodel (risk ≈ wP^T 1D) the expected final risk reduces to a weighted sum; the optimal B item teaching set is the B items with smallest weights wP,x (1−Px)^n (efficient selection).

Overall, the paper gives theoretical guarantees (optimality or approximation ratios), proof sketches, algorithms/pseudocode in the appendix, and synthetic experiments (linear classifiers, axis aligned rectangles, datamodel selection) showing improvements over no teach and a simulated random baseline.

**Strengths:**

•	Problem novelty and relevance: NtN formalizes a realistic two phase teaching scenario (limited guided teaching followed by natural i.i.d. learning).

•	Clear separation of settings: instance agnostic vs instance aware capture different practical knowledge regimes and motivate different techniques.

•	Theoretical contributions: nontrivial results for multiple hypothesis classes (exact VC reductions for linear class, approximation for finite classes, concrete budget→VC tables for rectangles).

•	Elegant reduction for instance aware case: datamodel linearization yields a simple, efficient selection rule with clear interpretation (weight scaled by (1−Px)^n).

•	Reproducibility support: proofs, pseudocode and experimental details (meta dataset construction, datamodel learning) are provided in the appendix.

•	Empirical validation: synthetic experiments verify expected behaviors (higher budget reduces final risk; proposed methods outperform naive baselines in the tested regimes).

**Weaknesses:**

•	Strong assumptions for instance aware solution: the linear datamodel assumption (risk exactly linear in dataset indicators) is strong; the paper lacks analysis of the effect of datamodel approximation/error on selection quality and final risk.

•	Limited scalability and realism of experiments: evaluations are on small, synthetic, low dimensional/discrete domains (e.g., 16 point circle, small grids). Methods that require enumerating version spaces or finding feature space preimages may not scale to high dimensional real data (images, large corpora).

•	Constructive/algorithmic gaps: for some claims (polynomial preimages, rectangle examples) the paper assumes existence or states constructions but provides limited practical algorithms or complexity analysis for finding these examples in constrained domains.

•	Strong learner assumptions: realizability and version space learner assumptions simplify analysis but reduce applicability under label noise, model misspecification, or when learners produce single hypotheses (not full VS).

•	Baselines: the simulated baseline (random simulated teaching sets) is weak; stronger heuristics (greedy VC reduction, information gain, uncertainty sampling) are not compared.

•	Computational cost of datamodel training: building the meta dataset requires training many base learners on many subsets; costs and required meta sample sizes are not discussed.

•	Lack of real world scenarios and reduced practical contribution: the paper provides only synthetic, small scale experiments and no demonstrations on real datasets or tasks. This reduces the perceived practical contribution and leaves unclear whether the methods (particularly datamodel learning and preimage constructions) work in realistic, high dimensional settings.

**Questions:**

Suggested reviewer questions for the authors
1.	Datamodel robustness: if the learned datamodel ŵ deviates from the true w (||ŵ − w|| large), can you bound how selection via ŵ affects the expected final risk? How accurate must the datamodel be in practice for the instance aware selection to be beneficial?

2.	Scalability and practicality: how do your algorithms scale when |X| is large and the hypothesis class or feature map is high dimensional? For the linear and polynomial constructions, how do you find the required vectors/preimages when only a constrained finite X is available?

3.	Relaxing realizability: how do your instance agnostic results change when realizability fails (label noise or h* not in H)? Can your VC reduction approach be adapted to agnostic or noisy settings?

4. Stronger baselines: have you compared OPT VC / OPT DM to greedy heuristics that approximate VC reduction or information gain selection? If not, can you run such comparisons?

5.	Datamodel training cost: how many meta subsets and base trainings are needed to obtain a usable datamodel in your synthetic experiments? Can you estimate computational requirements for larger problems and propose practical approximations?

6.	Feature preimage existence: for polynomial/kernel results you assume preimage existence. Which common kernels satisfy this, and what do you propose when preimages do not exist?

7.	Negative examples and reduction: the linear class analysis suggests negative labeled examples do not help reduce ambient dimensionality. Can you provide intuition or caveats when you cannot freely choose inputs (constrained X) or under noisy labels?

8.	Empirical sensitivity: can you show sensitivity analyses for OPT DM to (a) number of meta samples, (b) regularization λ in Lasso, and (c) errors in estimated PX (when PX is estimated rather than known)?

9.    Practical teaching scenarios (new suggestion): please consider evaluating or discussing more realistic practical teaching scenarios to demonstrate broader applicability. For example: Large pretrained models or LLMs teaching downstream agents (e.g., an LLM producing demonstrations or curricula for smaller RL agents or classifiers).

---

> ### Author Response · Authors · 2025-11-24
>
> We thank the reviewer for recognizing the "problem novelty," "theoretical contributions," and "elegant reduction" in our work. We appreciate the rigorous technical questions and address the concerns regarding assumptions, baselines, and practical scope below.
>
> ## 1. Linear Datamodel: Robustness, Sensitivity & Assumptions (Q1, Q5, Q8)
>
> You raised valid concerns regarding the robustness of the linear assumption ($Risk \approx w^T \mathbb{I}_D$) and the sensitivity of the method to estimation errors.
>
> - **Sensitivity Analysis (Response to Q8):** We have performed the requested sensitivity analyses for OPT-DM:
>
>     - **Meta-dataset size:** For a fixed teaching budget $B$, we plotted test risk vs. nature budget ($n_{iid}$) for various meta-dataset sizes ($500$ to $5000$) sampled from a uniform distribution. Performance is stable for sizes $\ge 1000$. Below this threshold, the approximation degrades. We will include these plots in the revision.
>
>     - **Regularization ($\lambda$):** We do not fix $\lambda$ apriori. As noted in Appendix C.2.1, we use `LassoLarsCV` with 4-fold cross-validation to automatically select the optimal $\lambda$ that provides the best $\ell_1$-regularized fit.
>
>     - **Errors in $P_X$:** Linear regression theory suggests $\lVert \hat{\theta} - \theta_{\text{true}} \rVert_2 \leq K \cdot d(P_X, P_{X, \text{est}})$. As long as the distribution $P_{true}$ is estimated well, the datamodel remains a reliable surrogate for risk.
>
> - **Tool vs. Application:** We reiterate that the Linear Datamodel is an off-the-shelf tool we adopt [Ilyas et al., 2022]. While analyzing linear datamodel for approximation error and misfit would be an intersting direction, this is independent to our contribution: utilizing function approximation to solve the instance-aware teaching objective. We have also provided a way forward in **Appendix C.3** to use non-linear Neural Datamodels (via Algorithm 4) when the linear assumption fails.
>
>
>
> ## 2. Relaxing Realizability & Agnostic Settings (Q3)
>
> Thanks for you question on how our results change when realizability fails (agnostic/noisy settings).
>
> - **VC Dimension Universality:** While our initial derivation uses realizability for tractability, the core mechanism - minimizing VC dimension - remains the optimal strategy in the agnostic setting. The teacher's goal is to reduce the complexity (VC dimension) of the set of plausible candidates, ensuring the class-optimal hypothesis $h^*$ survives while the complexity bound for learning in the nature phase shrinks.
>
> - **Theoretical Basis:** Standard PAC learning theory establishes that the VC dimension characterizes the sample complexity for _both_ realizable ($O(d/ \epsilon)$) and agnostic ($O(d/ \epsilon^2)$) settings. Therefore, our algorithm (OPT-VC) effectively minimizes the worst-case error bound regardless of the regime.
>
>
> ## 3. Baselines: Greedy Heuristics & Uncertainty Sampling (Q4)
>
> - **Intractability of Greedy VC:** We emphasize that for general hypothesis classes, calculating the VC dimension of a version space is NP-hard (Theorem 1 context). Therefore, a true "Greedy VC" algorithm is computationally intractable. Our "Simulated Teaching" baseline acts as the tractable Monte-Carlo approximation of this greedy search.
> - **Active Learning vs. Teaching:** Information Gain and Uncertainty Sampling are Active Learning (sequential query) metrics. In NtN, the teacher must construct a batch $D_T$ _a priori_. We will add a discussion contrasting our approach against these sequential AL heuristics.
>
> ## 4. Negative Examples & Dimensionality Reduction (Q7)
>
> When ideal positive vectors are absent in a constrained pool $X$, we can rely on the optimization framework (see Point 4) to identify the best available subset.
>
> Subspace Approximation: The optimization naturally selects vectors that are maximally orthogonal to $w^*$ within the constraints. These "near-orthogonal" vectors can still effectively constrain the version space along critical dimensions, achieving significant subspace reduction ("nearly killing the subspace") even without ideal orthogonal points.

---

> > ### Author Response · Authors · 2025-11-24
> >
> > ## 5. Computing Feature Preimages (Q6)
> >
> > We appreciate you asking a deeper level of question. Please note that linear case is a good example where pre-images exists. While this assumption has been made in prior works [Kumar et. al. 2021], we understand that it may not be easy to directly compute them for general kernel. To address that, we proposed a practical algorithm detailed in **Appendix B.4**, which solves an optimization problem to find a good dataset.
> >
> > When pre-images do not exist, one can still solve the optimization problem to find dataset that induce nearly orthogonal features. Our current solution solves the problem iteratively and can accumulate significant error. However, we suggest a reformulation(as stated below) to extend the optimization problem to jointly solve for all orthogonal components simultaneously.
> >
> > - **Main Idea:** Instead of finding teaching points sequentially (which risks error accumulation), we optimize all $B$ inputs $X = \{x_1, \dots, x_B\}$ simultaneously to satisfy the geometric requirements of Theorem 5 end-to-end.
> >
> > - Joint Objective: We minimize a unified differentiable loss $\mathcal{L}(X)$ that enforces orthogonality and the "zero-sum" closure constraint as soft penalties:
> >
> >     $$\min_X \sum_{i=1}^{B-1} (\phi(x_i)^\top w^*)^2 + \lambda_1 \sum_{i < j} (\phi(x_i)^\top \phi(x_j))^2 + \lambda_2 \left\|\sum_{k=1}^B \phi(x_k)\right\|^2$$
> >
> > - **Mechanism:** By minimizing $\mathcal{L}(X)$ via gradient descent, the algorithm finds a configuration where every element is a valid preimage that maximally satisfies the theoretical constraints, avoiding local minima.
> >
> >
> > ## 6. Experimental Realism & Scalability (Q2, Q9)
> >
> > We thank the reviewer for the forward-looking suggestion to evaluate on realistic scenarios like LLMs.
> >
> > - **Foundational Goal:** As this is the **first paper** to formalize and theoretically analyze the budget-constrained "Nurture-then-Nature" framework, we purposefully utilized simplified settings to provide rigorous groundwork. Our experiments serve as **empirical validations of the theory** rather than benchmarks for state-of-the-art engineering performance.
> >
> > - **The "Inaugural Paper" Scope:** We respectfully suggest that expecting a single inaugural paper to solve all practical issues and scale to complex real-world datasets immediately sets an incredibly high bar. We hope our work is evaluated in the correct light: as a **theoretical foundation** focused on providing efficient algorithms and provable guarantees, while outlining a clear path for future practical extensions via datamodels, rather than as an applied paper.
> >
> >
> > We hope these clarifications address your concerns and helps to evaluate our work in correct light. Should you have any follow up questions, please let us know.

---

### Official Review · Reviewer_HU2H · 2025-11-01

**Soundness:** 2
**Presentation:** 2
**Contribution:** 2
**Rating:** 2
**Confidence:** 2

**Summary:**

This paper introduces a new framework, "nurture-then-nature", which focuses on optimal dataset design for a first phase that comes from a teacher with a limited budget, and aims to minimize error after learning from a nature phase, whose distribution is either known or unknown by the teacher. It presents algorithms with guarantees for both settings, and provides experiments to help conceptually explain and support the theory.

**Strengths:**

- introduces and studies a new framework for budget-constrained teaching, inspired by practical situations
- provides experiments that aim to conceptually clarify the theoretical aspects
- i am less familiar with the theory, but sections 4 and 5 seem to make sense under the assumptions

**Weaknesses:**

- while there is practical motivation for the setting, there is less demonstrated practical applicability, especially in the experiments section
- the experiments are designed with toy datasets specifically with the framework in mind, but do not provide empirical justification for the strength of the algorithms introduced. this could potentially be improved with the use of real datasets and additional baselines
- assumptions made seem unlikely to hold in a realistic setting, which would require additional experiments to demonstrate practicality

**Questions:**

Most of my concerns are with the experimental section, as described in weaknesses. I would be happy to adjust my review if they are addressed, or if the authors could justify why the existing experiments are sufficient.

---

> ### Author Response · Authors · 2025-11-24
>
> We thank the reviewer for recognizing the novelty of the "Nurture-then-Nature" framework. We address the concerns regarding practical applicability and assumptions below.
>
> ## 1. Foundational Nature & Scope of Contribution
>
> We wish to emphasize that this is the first paper to formalize and theoretically analyze the budget-constrained "Nurture-then-Nature" framework.
>
> - **Theoretical Focus:** Like pioneering works in machine teaching [e.g., Goldman & Kearns, 1995; Zhu et al., 2018], we rely on simplified settings (realizability, linear models) to establish rigorous hardness results and sample complexity bounds. These standard assumptions are necessary starting points to verify that the NtN problem is solvable and to demonstrate how it differs from unconstrained teaching.
>
> - **Role of Experiments:** Consequently, our experiments are designed as **controlled verifications** of these theoretical bounds rather than benchmarks for state-of-the-art performance. In complex settings like deep learning, confounding factors (optimization noise, architectural bias) make it impossible to isolate the specific gains of the NtN strategy, whereas our controlled settings provide exact validation of the theory.
>
>
> ## 2. Expectations for Inaugural Work
>
> We respectfully submit that requiring a single inaugural paper to bridge the gap from theoretical inception to large-scale deployment sets an unrealistic standard for novel frameworks. We invite the reviewer to evaluate this work as a theoretical foundation: one that proves the solvability of the NtN problem, provide efficient algorithms and the necessary algorithmic blueprints (via datamodels) for future scaling, rather than as a final applied engineering solution.
>
>
> We hope these clarifications address your concerns. We are happy to answer any follow up questions that you may have.

---

### Author Response · Authors · 2025-11-24

We thank the reviewers (HU2H, RAFk, PaNq, G9gh) for their insightful feedback and for recognizing the "problem novelty", "theoretical contributions" (RAFk), and that the paper is "well written" (PaNq). We are encouraged that the reviewers found the Nurture-then-Nature (NtN) framework interesting and relevant.

We have addressed the common concerns raised across reviews as follows:

1. **Scope of Contribution (Foundational vs. Applied):** A recurring theme was the use of simplified experimental settings. We would like emphasize that this is a foundational  **theoretical work** in a new direction of study. Our primary goal has been to provide efficient algorithms with provable guarantees and concrete algorithmic pathways (using datamodels) for future real world extensions. The experiments are designed as **controlled verifications** of these algorithmic results and their guarantees rather than as benchmarks for LLM/Deep Learning performance.

2. **Robustness & Sensitivity:** To address concerns about the "Instance-Aware" assumptions (RAFk), we will add **sensitivity analyses** demonstrating that our selection mechanism is robust to estimation errors in the Datamodel.

3. **Algorithmic Improvements:** We will upgrade the preimage discovery algorithm in Appendix B.4 to a **Joint Optimization** approach (soft orthogonality constraints), ensuring more robust solution finding in constrained feature spaces.

4. **Clarifications:** We will expand the discussion on baselines, clarified the definition of "Instance-Awareness" (G9gh), and corrected the template title oversight (PaNq).


We believe these revisions strengthen the paper's position as a rigorous first step in this new direction.

---

### Meta-Review · Area_Chair_auxK · 2026-01-07

**Summary:**

The paper proposes a two-phase machine teaching setting where a budget-limited teacher provides a small curated dataset (nurture) before the learner continues with i.i.d. data from the environment (nature).

Across reviews, there is broad agreement that:
* Problem setting is novel and practically motivated
* The paper contains nontrivial theory, especially for the instance-agnostic regime via VC-based reductions and for several concrete hypothesis classes (finite classes, rectangles, linear separators, polynomial features).

At the same time, there are also shared concerns
* Empirical support is limited: experiments are small-scale and synthetic, and do not convincingly establish practical utility beyond conceptual validation (HU2H, RAFk, G9gh).
* Strong modeling assumptions for the key claims (realizability, version-space learners in the instance-agnostic part; exact linear-datamodel realizability and strong knowledge assumptions in the instance-aware part), and the paper lacks a clear robustness story when these assumptions are violated (RAFk, HU2H).
* Positioning issues: one reviewer flags that optimality in the agnostic setting is with respect to a VC-based upper bound, not true post-nature error, and asks for clearer positioning against curriculum/active learning/data selection literatures (PaNq, RAFk).
* The computational cost of datamodel training are not convincingly addressed for more complex domains (RAFk).

The authors' rebuttal clarifies that the work is intended as foundational learning theory, commits to additional sensitivity analyses (datamodel stability), proposes a more robust joint optimization for the preimage step along with further clarifications in presentation. These help, but do not fully resolve the core concerns around empirical support, robustness, and practical scalability for the current submission.

**Reviewer Concerns:**

Concerns addressed
* The major additional results are on the datamodel sensitivity checks. Authors state they have run sensitivity analyses (meta-dataset size, regularization via CV, errors in and will include plots; this partially addresses RAFk's concern on empirical sensitivity.
* Clarity regarding positioning and presentation should be addressed

Concerns partially addressed or outstanding:
* the rebuttal largely defends the use of toy experiments as proof of concept rather than adding stronger empirical evidence
* robustness under violated assumptions is not thoroughly justified
* proposing a joint optimization for preimages is a reasonable direction, but it is not yet demonstrated or analyzed

**Reviewer Scores:**

Even with modest upward movement for one or two reviewers, I do not expect the discussion to converge to a clear accept.

---

### Decision · Program_Chairs · 2026-01-26

Reject